# IL-13Rα2 Is Involved in Resistance to Doxorubicin and Survival of Osteosarcoma Patients

**DOI:** 10.3390/ph17111526

**Published:** 2024-11-13

**Authors:** Maryam Karamikheirabad, Junyue Zhang, Ae-Ri Ahn, Ho Sung Park, See-Hyoung Park, Young Jae Moon, Kyoung Min Kim, Kyu Yun Jang

**Affiliations:** 1Department of Pathology, Medical School, Jeonbuk National University, Jeonju 54896, Republic of Korea; marya.karami@gmail.com (M.K.); yuezai123@naver.com (J.Z.); arahn@jbnu.ac.kr (A.-R.A.); hspark@jbnu.ac.kr (H.S.P.); 2Department of Bio and Chemical Engineering, Hongik University, Sejong 30016, Republic of Korea; shpark74@hongik.ac.kr; 3Department of Biochemistry and Molecular Biology, Medical School, Jeonbuk National University, Jeonju 54896, Republic of Korea; yjmoonos@jbnu.ac.kr; 4Department of Orthopedic Surgery, Jeonbuk National University Hospital, Jeonju 54907, Republic of Korea; 5Research Institute of Clinical Medicine, Jeonbuk National University, Jeonju 54896, Republic of Korea; 6Biomedical Research Institute, Jeonbuk National University Hospital, Jeonju 54907, Republic of Korea

**Keywords:** osteosarcoma, IL-13Rα2, prognosis, doxorubicin

## Abstract

Background/Objectives: Interleukin 13 receptor alpha 2 (IL-13Rα2) is a receptor with a high affinity for IL-13 and is involved in the progression of human cancers. However, studies on the role of IL-13Rα2 in osteosarcoma are limited. Therefore, this study aimed to investigate the expression and roles of IL-13Rα2 in the progression of osteosarcoma. Methods: This study evaluated the roles of IL-13Rα2 in osteosarcomas by evaluating tumor tissues from 37 human osteosarcomas and osteosarcoma cells. Results: Immunohistochemical positivity of IL-13Rα2 was an independent indicator of shorter overall survival and relapse-free survival of 37 osteosarcoma patients and 26 subpopulations of patients who received adjuvant chemotherapy with multivariate analysis. In U2OS and KHOS/NP osteosarcoma cells, overexpression of IL-13Rα2 significantly increased proliferation, migration, and invasion of cells, all of which decreased with knockdown of IL-13Rα2. Overexpression of IL-13Rα2 increased expression of TGF-β, snail, cyclin D1, and BCL2 but decreased BAX, and knockdown of IL-13Rα2 caused a decrease in expression of these molecules. In addition, both in vitro and in vivo, proliferation of osteosarcoma cells increased, and apoptosis decreased with overexpression of IL-13Rα2 under treatment with doxorubicin. Knockdown of IL-13Rα2 sensitized osteosarcoma cells to the cytotoxic effect of doxorubicin. Conclusions: The results of this study suggest that the expression of IL13Rα2 might be used as a potential prognostic indicator in osteosarcoma patients. Furthermore, it is observed that IL13Rα2 influences the resistance to the chemotherapeutic agent doxorubicin. Therefore, a therapeutic trial targeting IL13Rα2 might be a new therapeutic strategy for osteosarcoma, especially those highly expressing IL13Rα2.

## 1. Introduction

The receptors of interleukin 13 (IL-13R) consist of two subunits, IL-13Rα1 and IL-13Rα2 [1,2]. Among these, IL-13Rα1 forms a heterodimer with IL-4Rα, consisting of a type II IL4 receptor [3], in which they stabilize each other and are involved in various biological pathogenesis, including cancer [4]. However, the binding affinity between IL-13 and IL-13Rα1 is relatively weak [5]. On the other hand, IL-13Rα2 has a strong affinity for IL-13 [6], and its activation is associated with the pathogenesis of various diseases, such as inflammatory bowel disease [7], dermatitis [8], and lung fibrosis [9]. In addition, higher expression of IL-13Rα2 has also been reported in several types of cancer, including glioblastoma [10], ovarian carcinoma [11], head and neck cancer [12], melanoma [13], and colorectal cancer [14]. Moreover, elevated IL-13Rα2 expression correlated with decreased survival rates in glioblastoma [15] and renal cell carcinoma patients [16] and a high risk of brain metastasis from breast cancer [17]. The prognostic significance of IL-13Rα2 expression in human cancers is associated with its involvement in various oncogenic pathways [16,18,19]. In renal cell carcinomas, IL-13Rα2 regulated cancer progression through the regulation of the JAK2/FOXO3 pathway [16]. In colorectal and breast cancer, IL-13Rα2 stimulated the invasiveness through the Src-PI3K pathway [20] and protein tyrosine phosphatase-1B [18]. IL-13Rα2 stimulated cancer progression in glioblastoma by activating the AP-1 pathway [19]. Furthermore, IL-13Rα2-mediated regulation of TGF-β expression in a lung fibrosis model [21] suggests that IL-13Rα2 may play a role in the invasion and metastatic potential of cancers through the epithelial-to-mesenchymal transition (EMT) process [14,21]. Therefore, it has been suggested that IL-13Rα2 could serve as a potential therapeutic target for malignant tumors.

Osteosarcoma is a leading malignant bone tumor primarily affecting young individuals [22]. Even though osteosarcoma accounts for approximately 1% of all malignancies [23], it is one of the most challenging to cure due to its aggressiveness and resistance to standard therapies [22]. Despite advances in the treatment of human cancers, the survival rate of advanced osteosarcoma patients remains low [22,24]. More than 20% of osteosarcoma patients have a survival rate of fewer than five years [25]. In addition to standard operation, radiation therapy [26] and doxorubicin-based chemotherapy [27] have shown therapeutic efficacy in osteosarcoma; however, resistance to these conventional anti-cancer therapies and therapy-related toxicity remains a significant challenge [28,29]. Therefore, new therapeutic strategies for osteosarcoma need to be developed [30]. Recently, immunotherapeutic approaches, including immune checkpoint inhibitors, tumor microenvironment modulators, cytokine therapy, and chimeric antigen receptor T-cell therapy, are being investigated for osteosarcoma treatment [31,32]. Considering the role of IL-13Rα2 in cancer progression [16,33] and the effectiveness of blocking IL-13Rα2-related pathways in inhibiting and deteriorating cancer cells [33,34], IL-13Rα2 might be a potential therapeutic target for osteosarcoma. In addition, IL-13 is involved in the regulation of osteoblast and osteoclast activation [35,36], and IL-13Rα2 regulates osteoclastic differentiation through the mitogen-activated protein kinase/Akt pathway [37]. Therefore, considering the potential roles of the IL-13/IL-13Rα2 pathway in osteogenesis and its roles in the development and progression of osteosarcoma, this study aimed to explore the expression and roles of IL-13Rα2 in the progression of osteosarcoma by evaluating the expression of IL-13Rα2 in human osteosarcoma tissue samples and cell lines.

## 2. Results

### 2.1. Higher Expression of IL-13Rα2 Is Linked to Reduced Survival in Osteosarcoma Patients

In osteosarcoma tissues, immunohistochemical staining revealed IL-13Rα2 in both the cytoplasm and nuclei of osteosarcoma cells (Figure 1A). IL-13Rα2 expression positivity was assessed using a receiver operating characteristic curve, establishing a cut-off immunohistochemical staining score of 14 (area under the curve: 0.810; *p* = 0.001; 95% CI [95% confidence interval]: 0.662–0.958) (Figure 1B). Cases scoring 14 or higher were classified as IL-13Rα2-positive (Figure 1B). At this cut-off value, IL-13Rα2 positivity was only significantly associated with sex (*p* = 0.014) (Table 1). However, in survival analysis, higher expression of IL-13Rα2 was significantly associated with shorter OS (HR [hazard ratio]: 7.191; 95% CI: 2.063–25.067; *p* = 0.002) and RFS (HR: 5.668; 95% CI: 1.848–17.385; *p* = 0.002) in univariate analysis (Table 2) (Figure 1C). In addition, univariate analysis indicated that age, tumor size, tumor stage, T category, and M category had significant associations with OS or RFS (Table 2). A multivariate analysis was conducted, including age, tumor size, stage, T and M categories, and IL-13Rα2 expression, all of which were significantly associated with OS or RFS in the univariate analysis. Multivariate analysis identified IL-13Rα2 positivity as an independent prognostic factor for OS and RFS (Table 3). Positivity for IL-13Rα2 was associated with a 7.191-fold increased risk of mortality in osteosarcoma patients (95% CI: 2.063–25.067; *p* = 0.002) and a 6.616-fold increased likelihood of tumor relapse or patient death (95% CI: 1.909–19.883; *p* = 0.002) (Table 3). Age (*p* = 0.014) and tumor stage (*p* = 0.027) were the independent indicators of RFS in multivariate analysis (Table 3).

Furthermore, in this study, despite the prognostic significance of IL-13Rα2 expression, there was no clinical prognostic factor that was significantly associated with IL-13Rα2 expression (Table 1). Therefore, we hypothesized that IL-13Rα2 might be associated with therapeutic efficacy for the conventional treatment of osteosarcoma, and we further analyzed the 26 patients who had received postoperative chemotherapy. In this subpopulation, IL-13Rα2 positivity was significantly associated with OS and RFS (Figure 2). Furthermore, IL-13Rα2 positivity was an independent indicator of shorter OS (HR: 11.985; 95% CI: 2.600–55.254; *p* = 0.001) and RFS (HR: 7.461; 95% CI: 1.946–28.605; *p* = 0.003) for osteosarcoma patients who had received adjuvant chemotherapy in multivariate analysis (Table 4).

### 2.2. IL-13Rα2 Is Involved in the Proliferation and Invasion Activity of Osteosarcoma Cells

Based on the role of IL-13Rα2 expression in the prognosis of osteosarcoma, the effects of IL-13Rα2 on the proliferation of osteosarcoma cells were evaluated. In both U2OS and KHOS/NP cells, IL-13Rα2 overexpression promoted cell proliferation, while knockdown of IL-13Rα2 led to decreased cell growth, as shown through the CCK-8 assay (Figure 3A). The colony-forming assay also showed that overexpression of IL-13Rα2 significantly increased the colony-forming ability of cells, and knockdown of IL-13Rα2 significantly decreased colony-forming ability in both U2OS and KHOS/NP cells (Figure 3B).

In addition, in the trans-chamber migration and invasion assay, the number of migrated or invaded cells significantly increased with overexpression of IL-13Rα2 and significantly decreased with knockdown of IL-13Rα2 (Figure 4A,B). In line with the effect of IL-13Rα2 on osteosarcoma cell growth and invasion, it also plays a role in signaling pathways related to cell proliferation, invasion, and apoptosis. Knockdown of IL-13Rα2 reduced protein and mRNA levels of cyclin D1, snail, TGF-β, and BCL2, while increasing BAX expression in osteosarcoma cells (Figure 5A,B). Conversely, IL-13Rα2 overexpression increased protein and mRNA levels of cyclin D1, snail, TGF-β, and BCL2 and reduced BAX levels in both cell types (Figure 5A,B).

### 2.3. IL-13Rα2 Is Involved in Resistance to the Cytotoxic Effect of Doxorubicin and Silencing of IL-13Rα2 Sensitized Osteosarcoma Cells to Doxorubicin

In osteosarcoma patients who received adjuvant chemotherapy, higher expression of IL-13Rα2 predicted shorter survival. In addition, overexpression of IL-13Rα2 in osteosarcoma cells inhibited pre-apoptotic signaling. Therefore, based on these results, the impact of IL-13Rα2 on osteosarcoma cell survival during doxorubicin treatment was further evaluated. In a CCK-8 assay, the cellular proliferation of U2OS and KHOS/NP cells induced to overexpress IL-13Rα2 was higher compared with control cells under doxorubicin treatment (Figure 6A). Osteosarcoma cells with a knockdown of IL-13Rα2 were more sensitive to doxorubicin-mediated cytotoxicity in a CCK-8 assay (Figure 6A). The colony-forming ability of cells under treatment with doxorubicin was significantly higher with overexpression of IL-13Rα2 and significantly lower with knockdown of IL-13Rα2 in both U2OS and KHOS/NP cells (Figure 6B). In a tumorigenic mice model, knockdown of IL-13Rα2 significantly decreased tumor growth under treatment with doxorubicin in KHOS/NP cells (Figure 6C). In contrast, overexpression of IL-13Rα2 significantly increased tumor growth under treatment with doxorubicin (Figure 6C).

Furthermore, in mice, the overexpression of IL-13Rα2 counteracted the tumor growth inhibitory effect of doxorubicin, resulting in no significant difference in tumor size between the IL-13Rα2 overexpression group and the control group not treated with doxorubicin (Figure 6C). Under treatment of doxorubicin, the knockdown of IL-13Rα2 significantly increased the expression of the protein of FOXO3a, BAX, and cleaved caspase 3, while reducing BCL2 expression (Figure 7A). Conversely, IL-13Rα2 overexpression significantly lowered FOXO3a, BAX, and cleaved caspase 3 levels but increased BCL2 expression under doxorubicin treatment (Figure 7A). Flow cytometric apoptotic analysis with Annexin V showed that apoptotic cells were significantly increased with knockdown of IL-13Rα2 under treatment with doxorubicin but significantly decreased with overexpression of IL-13Rα2 under treatment with doxorubicin in both U2OS and KHOS/NP cells (Figure 7B).

## 3. Discussion

This study showed that elevated IL-13Rα2 expression correlates with reduced survival rates in osteosarcoma patients. Univariate analysis showed that osteosarcoma patients with higher IL-13Rα2 expression have significantly shorter OS and RFS. This association remains significant even after adjusted in multivariate analysis with other clinical factors such as age, tumor size, stage, and metastatic status, suggesting that IL-13Rα2 is an independent prognostic indicator in osteosarcoma. Consistent with our findings, increased immunohistochemical IL-13Rα2 expression was significantly associated with poorer survival outcomes in patients with gastric carcinoma [38], renal cell carcinoma [16], colorectal cancers [14], and luminal-type breast carcinomas [39]. Furthermore, elevated IL-13Rα2 mRNA expression was significantly associated with reduced survival in glioblastoma [40] and adrenocortical carcinoma [41]. Therefore, it is suggested that IL-13Rα2 expression might be used as a prognostic indicator of human cancers, including osteosarcomas. Identifying patients with high IL-13Rα2 expression might be helpful in stratifying those at higher risk of poor outcomes to undergo new therapeutic regimens. Consequently, the implications of these findings might be helpful in the management of osteosarcoma patients.

Considering the prognostic impact of IL-13Rα2 in osteosarcoma, it is suggested that IL-13Rα2 expression might be associated with molecular mechanisms of cancer progression. In this study, beyond the prognostic relevance of IL-13Rα2 expression in osteosarcoma patients, IL-13Rα2 expression showed a close association with the proliferation and invasion activity of osteosarcoma cells. Overexpression of IL-13Rα2 significantly increased osteosarcoma cell proliferation, migration, and invasion, while knockdown of IL-13Rα2 suppressed these activities. These effects of IL-13Rα2 on osteosarcoma cells were associated with the expressions of molecules related to proliferation and epithelial-to-mesenchymal transition (EMT). Overexpression of IL-13Rα2 stimulated expression of TGF-β and snail, and knockdown of IL-13Rα2 downregulated TGF-β and snail. Given the role of TGF-β in EMT activation through the regulation of snail and ZEB gene expression [42,43], the increased invasiveness of IL-13Rα2-overexpressing osteosarcoma cells might be correlated to the EMT phenotype of cells. Supportively, it has been reported that IL-13 induces TGF-β expression, and silencing of IL-13Rα2 inhibited the expression of TGF-β [21]. In addition, in colorectal cancer cells, higher expression of IL-13Rα2 increased invasion activity through activation of Src, and silencing of IL-13Rα2 decreased hepatic metastasis of cancer cells in an in vivo model [14]. Consistently, IL-13Rα2 stimulated in vivo metastasis of pancreatic cancer cells in animal models [44]. In IL-13Rα2-targeted therapy, blocking IL-13Rα2 with monoclonal antibodies inhibited hepatic metastasis of colorectal cancer cells in animals through suppression of the Src-PI3K/AKT pathway [45]. Therefore, it is suggested that IL-13Rα2 might be important in the metastatic potential of cancer cells, and advanced cancer with higher expression of IL-13Rα2 might benefit from new therapies based on blocking the IL-13Rα2 pathway.

Besides its roles in osteosarcoma cell proliferation and invasiveness, IL-13Rα2 also plays a role in the regulation of signaling pathways related to cell survival and apoptosis. In this study, the knockdown of IL-13Rα2 decreased the expression of BCL2, antiapoptotic signaling, while increasing the expression of the pro-apoptotic protein BAX. Conversely, overexpression of IL-13Rα2 promotes cell survival by enhancing antiapoptotic signaling and suppressing pro-apoptotic pathways. These findings suggest that IL-13Rα2 might be contributing to the resistance of osteosarcoma cells to apoptosis. Supportively, the IL-13 pathway has been implicated in treatment resistance in several types of cancers. In NK/T-cell lymphoma cells, IL-13 induced resistance to Adriamycin by promoting the expression of ATP-binding cassette subfamily C member 4 [46]. Additionally, IL-4 and IL-13 from group 2 innate lymphoid cells suppressed antitumor immunity in a colorectal cancer model [47], and the production of IL-13 by group 2 innate lymphoid cells was upregulated by melanoma cells in vitro [48]. These findings suggest that a cross-talk between tumor cells and IL-13 production by innate lymphoid cells contributes to cancer progression. Moreover, blocking immune-suppressor-inducing molecules, such as netrin-1, suppressed resistance to chemotherapy and immune checkpoint inhibitors in a breast cancer model [49]. Higher expression of IL-13Rα2 was also associated with castration resistance in prostate cancer cells [50]. Therefore, in this context, it is suggested that IL-13 induction might be a common mechanism of tumor resistance, with IL-13Rα2 contributing to resistance as the receptor for IL-13.

Considering the therapeutic implication of overcoming resistance to anticancer cytotoxic agents of osteosarcomas, the impact of IL-13Rα2 knockdown or overexpression on cell proliferation and apoptosis in response to doxorubicin treatment was evaluated. In this context, IL-13Rα2 contributes to the resistance to doxorubicin, a commonly used chemotherapeutic agent in osteosarcomas. Doxorubicin was less effective in osteosarcoma cells overexpressing IL-13Rα2. IL-13Rα2-overexpressing osteosarcoma cells were resistant to doxorubicin, as evidenced by higher cell proliferation and the colony-forming ability of cells and in vivo tumor growth in mice under treatment with the drug. The resistance to doxorubicin was associated with the modulation of apoptotic pathways. IL-13Rα2 overexpression was associated with decreased expression of pro-apoptotic factors, FOXO3a, BAX, and cleaved caspase 3, while the expression of the antiapoptotic protein BCL2 was increased. In contrast, the knockdown of IL-13Rα2 sensitized the cells to doxorubicin, increasing apoptosis and reducing tumor growth in mice. Therefore, the shift towards an antiapoptotic state by IL-13Rα2 overexpression likely underlies the resistance to doxorubicin-induced cell death in osteosarcoma. Consistently, activation of FOXO induced apoptosis of cells and suppressed new tumor growth [51], and the DNA damaging agent mitoxantrone induced apoptosis of U2OS and MG63 osteosarcoma cells via activation of FOXO3a [52]. In renal cell carcinoma cells, knockdown of IL-13Rα2 or inhibition of IL-13Rα2 with telmisartan increased FOXO3 expression and induced apoptosis of cancer cells [16]. Similarly, the resistance of osteosarcoma cells to genotoxic agents is reported to be mediated by various molecules related to DNA damage repair to escape from apoptosis [53]. SIRT6 induced doxorubicin resistance through inhibition of apoptosis by activating a DNA damage repair pathway [53]. Therefore, these results suggest that IL-13Rα2 might serve as a viable target to counteract chemoresistance in osteosarcoma. Nonetheless, research on IL-13Rα2 in sarcomas, particularly osteosarcoma, remains limited. Among sarcoma types, higher expression of IL-13Rα2 in malignant peripheral nerve sheath tumors has been reported, and a therapeutic approach targeting IL-13Rα2 showed suppression of tumor growth and elongation of survival time in a murine tumor model [54]. Therefore, despite limited reports on the therapeutic efficacy of IL-13Rα2 in sarcomas, inhibition of IL-13Rα2 might enhance the efficacy of doxorubicin and potentially other chemotherapeutic agents, thereby improving outcomes for osteosarcoma patients with high IL-13Rα2 expression.

Recently, IL-13Rα2, as a cytokine receptor, has been employed as an anticancer immunotherapy, especially in glioblastoma and melanoma [55]. Strategies using immunotoxin, monoclonal antibodies, chimeric antigen receptor T-cells, tumor vaccine, and peptide–drug conjugates targeting have been proposed and are under extensive investigation [55]. In particular, monoclonal antibodies specific for IL-13Rα2, which disrupt IL-13/IL-13Rα2 interactions, improved the survival of mice with orthotopic glioma xenografts [56] and inhibited liver metastasis of colorectal cancer cells expressing IL13Rα2 [45]. Moreover, there are ongoing clinical trials using IL-13Rα2-targeted chimeric antigen receptor T-cell therapy [55]. Therefore, osteosarcoma patients with shorter survival with elevated expression of IL-13Rα2 might benefit from IL-13Rα2-targeted immunotherapy. In addition, when considering the role of IL-13Rα2 in bone homeostasis and its dysregulation in osteosarcoma, comparing it with non-cancerous cells is essential to distinguish tumor-specific dysregulation from normal physiological processes. This distinction is crucial to employ IL-13Rα2 as a therapeutic target of osteosarcoma. However, this study has a limitation in that we did not investigate the role of IL-13Rα2 in normal control osteoblastic cells. Therefore, further study is needed to provide valuable insights into whether IL-13Rα2 represents a viable, specific therapeutic target for osteosarcoma, rather than solely a prognostic marker.

## 4. Materials and Methods

### 4.1. Osteosarcoma Patients and Tissue Samples

IL-13Rα2 expression was analyzed in osteosarcoma tissue samples collected from patients who underwent operations between January 1998 and December 2012 at Jeonbuk National University Hospital. This study included 37 cases with complete medical histories, histologic slides, and tissue blocks. Each case was reviewed using the latest World Health Organization classification of bone tumors [22]. Clinicopathological factors analyzed included patient age, sex, tumor size, tumor stage, and T, N, and M staging categories. None of the patients received neoadjuvant therapy, though postoperatively, 26 patients underwent adjuvant chemotherapy, 10 received radiotherapy, and 7 received both treatments. This study was approved by the Jeonbuk National University Hospital Institutional Review Board (CUH2023-08-061, approval date: 6 September 2023).

### 4.2. Immunohistochemical Staining and Scoring of Tissue Microarrays

IL-13Rα2 expression in human osteosarcoma tissue was assessed through immunohistochemical staining on tissue microarray (TMA) sections. Each case had two 3.0 mm cores from the region with the highest histologic grade of tumor cells from the original tissue blocks. The TMA tissue sections were deparaffined, and antigen retrieval was performed by boiling for 20 min in a pH 6.0 antigen retrieval buffer (DAKO, Glostrup, Denmark) using a microwave oven. The sections were then incubated with the antibody for IL-13Rα2 (#sc-134363, 1:100, Santa Cruz Biotechnology, Santa Cruz, CA, USA), developed with chromogen (DAKO, Glostrup, Denmark), and counterstained with hematoxylin. Two pathologists (KYJ and HSP) examined the stained slides using multiviewing microscopy, without access to clinical information, to reach a consensus. Immunohistochemical staining scores were calculated by adding a staining intensity scale (0: no staining; 1: weak intensity; 2: intermediate intensity; and 3: strong intensity) and a staining area scale (0: no stained cells; 1: 1%; 2: 2~10%; 3: 11~33%; 4: 34~66%; and 5: 67~100%) for each TMA core [57,58]. The final staining score, ranging from zero to sixteen, was obtained by summing the scores from each TMA core.

### 4.3. Cell Lines, Reagents, and Transfection

In this study, human osteosarcoma cell lines U2OS (Korean Cell Line Bank, Seoul, Republic of Korea) and KHOS/NP (generously provided by Chang-Bae Kong, Department of Orthopedic Surgery, Korea Institute of Radiological and Medical Sciences, Seoul, Republic of Korea) were utilized. U2OS and KHOS/NP cells were cultured in Dulbecco’s Modified Eagle Medium (DMEM) with 10% fetal bovine serum and 1% penicillin–streptomycin (100 units/mL of penicillin and 100 µg/mL of streptomycin). To induce knockdown or overexpression of IL-13Rα2, plasmids for short hairpin RNA (shRNA) of IL-13Rα2 (#sc-63339-SH, Santa Cruz Biotechnology, Santa Cruz, CA, USA) and human untagged clone DNA constructs for IL-13Rα2 (#SC125642, OriGene Technologies, Beijing, China) were transfected to osteosarcoma cells using Lipofectamine 3000 (#L3000015, Invitrogen, Carlsbad, CA, USA). Doxorubicin hydrochloride was purchased from Sigma-Aldrich (#D1515, Sigma-Aldrich, St. Louis, MO, USA).

### 4.4. Cell Proliferation and Colony-Forming Assays

Cell proliferation was assessed using Cell Counting Kit-8 (CCK-8, #CK04, Dojindo Molecular Technologies, Kumamoto, Japan) and colony-forming assay. For the CCK-8 assay, 3 × 10^3^ KHOS/NP and U2OS cells were seeded in 96-well plates and incubated for 24, 48, and 72 h, measuring absorbance at 450 nm with a microplate reader (Bio-Rad Model 680, Bio-Rad Laboratories Inc., Tokyo, Japan). Colony formation was analyzed by seeding 3 × 10^3^ U2OS cells and 1 × 10^3^ KHOS/NP in 6- or 12-well plates. The culture plates were fixed and stained with 0.5% crystal violet (#SLBN6236V, Sigma Aldrich, St. Louis, MO, USA), and colonies were quantified with ColonyCountJ software, an add-on program to the Fiji Image Software package (NIH, version 1.44 m, Bethesda, MD, USA) [59].

### 4.5. Migration and Invasion Assay

To assess cell migration, Transwell Boyden inserts (#353097, Falcon, Franklin Lakes, NJ, USA) were used. For the migration assay, 1 × 10^5^ cells were seeded in the upper chamber with a serum-free medium. Invasion assay was performed by seeding 2 × 10^5^ cells on the upper chamber of Matrigel-coated invasion chambers (#354480, Corning, Bedford, MA, USA). The lower chamber of the migration and invasion assay was filled with 500 μL DMEM with 10% FBS as a chemoattractant and incubated for sixteen hours. Cells that migrated or invaded to the lower side of the membrane were stained with Diff Quick stain solution (#468.037.1A, Diff-Quick, Dade Behring Inc., Newark, DE, USA). The number of migrated or invaded cells was counted in five × 200 microscopic fields for each case.

### 4.6. Western Blot

Protein lysate was obtained using a PRO-PREP™ solution (#17081, iNtRON Biotechnology, Seongnam, Republic of Korea). Proteins were separated by gel electrophoresis and transferred to a polyvinylidene difluoride membrane [60,61]. The following antibodies were used in Western blotting: IL-13RA2/CD213a2 (E7U7B) (#85677, Cell Signaling Technology, Beverly, MA, USA), transforming growth factor β (TGF-β) (#3709, Cell Signaling Technology, Beverly, MA, USA), snail (#ab180714, Abcam, Cambridge, UK), cyclin D1 (#2922, Cell Signaling Technology, Beverly, MA, USA), BAX (#2774, Cell Signaling Technology, Beverly, MA, USA), BCL2 (D17C4) (#3498, Cell Signaling Technology, Beverly, MA, USA), forkhead box O3 (FOXO3a) (75D8) (#2497, Cell Signaling Technology, Beverly, MA, USA), cleaved caspase-3 (Asp175) (5A1E) (#9664, Cell Signaling Technology, Beverly, MA, USA), and actin (#sc-47778, Santa Cruz Biotechnology, Santa Cruz, CA, USA). Proteins were detected on the membranes via enhanced chemiluminescence (ECL) reagents (#WBKLS0500, Immobilon Western HRP Substrate, Millipore, Darmstadt, Germany) and visualized by using a LAS-3000 Imaging System (LAS-3000, Fujifilm, Tokyo, Japan).

### 4.7. Flow Cytometric Analysis for Apoptosis

Apoptosis was evaluated with an Annexin V-FITC Apoptosis Detection Kit (#ab14085, Abcam, Waltham, MA, USA) by flow cytometry. U2OS (1 × 10^5^) and KHOS/NP (1 × 10^5^) cells were resuspended in 500 μL binding buffer and incubated with 5 µL Annexin V and 5 µL propidium iodide in a 5 mL polystyrene tube (Falcon™; Becton Dickinson, Franklin Lakes, NJ, USA) according to the manufacturer’s protocol. The tubes were then analyzed on a Becton Dickinson FACSCalibur Flow Cytometer (BD FACSCalibur, BD Bioscience, San Diego, CA, USA), and data were processed with FlowJo software (version 10.3, BD Life Sciences, Ashland, OR, USA).

### 4.8. Quantitative Reverse-Transcription Polymerase Chain Reaction

Total RNA was isolated using an RNeasy Mini Kit (#74004, Qiagen, GmbH, Hilden, Germany), and cDNA was synthesized with the ReverTra Ace qPCR RT Kit (#FSQ-101, Toyobo, Tokyo, Japan). A thermo-cycler (Applied Biosystems, A28132 QuantStudio 3 qPCR System, Life Technologies Holdings Pte. Ltd., Singapore) and the SYBR^®^ Green Realtime PCR Master Mix kit (#QPK-201, Toyobo, Tokyo, Japan) were used in the quantitative polymerase chain reaction. A quantitative polymerase chain reaction was performed using an ABI Prism 7900HT Sequence Detection System (Applied Biosystems, Foster City, CA, USA). The mRNA levels were normalized against GAPDH reference gene levels and calculated using the threshold cycle number method to have relative fold change for each target gene. The sequences of the primers used in this study for the reverse-transcription polymerase chain reaction are listed in Table 5.

### 4.9. In Vivo Tumorigenic Assay

The Institutional Animal Care and Use Committee of Jeonbuk National University approved the use of mice for this study (JBNU NON2024-003, approval date: 4 March 2024). A tumorigenic assay was performed using four-week-old male BALB/c nude mice (Orient Bio, Seongnam, Republic of Korea). Tumors were induced by injecting 2.5 × 10^6^ KHOS/NP cells transfected with empty vector, shRNA for IL-13Rα2, or plasmid for wild-type IL-13Rα2 into the back of mice subcutaneously. Three mice were used in each group. Doxorubicin (4 mg/kg in dimethyl sulfoxide) was administered intraperitoneally once per week, starting three weeks after tumor implantation. Tumor volume measurements were taken weekly and calculated using the formula: length × width × height × 0.52. Two weeks after the treatment with doxorubicin, animals were euthanized with carbon dioxide after sodium pentobarbital anesthesia to follow humane endpoints.

### 4.10. Statistical Analysis

Positivity for IL-13Rα2 immunohistochemical staining was used in receiver operating characteristic curve analysis to predict cancer death in osteosarcoma patients [62]. The cut-off point was determined as the highest positive likelihood ratio for the death of patients from osteosarcoma. Survival analysis focused on overall survival (OS) and relapse-free survival (RFS), with June 2015 as the study endpoint. For OS analysis, an event was defined as death due to osteosarcoma, while in RFS, events included osteosarcoma recurrence or death from the disease. OS and RFS durations were calculated from the surgery date to the event or last follow-up date. Statistical analysis included univariate and multivariate Cox proportional hazards regression and Kaplan–Meier survival analysis. Associations between factors were assessed using an ANOVA test, a chi-squared test, or a Student’s *t*-test. Each experiment was repeated three times, and representative results were presented. SPSS software (version 25.0, IBM, Armonk, NY, USA) was used for statistical evaluation, with *p*-values below 0.05 considered significant.

## 5. Conclusions

In conclusion, findings from this study indicate that IL-13Rα2 could serve as a novel marker of poor prognosis in osteosarcoma patients. In addition, the expression of IL-13Rα2 was associated with osteosarcoma cell proliferation and invasiveness. Furthermore, overexpression of IL-13Rα2 induced resistance to doxorubicin, and the knockdown of IL-13Rα2 sensitized osteosarcoma cells to doxorubicin. Considering that IL-13Rα2 is associated with reduced survival in osteosarcoma patients who received adjuvant chemotherapy, these findings suggest a potential role for IL-13Rα2 in resistance to standard anticancer treatments. Therefore, patients with osteosarcoma, particularly osteosarcomas with high IL-13Rα2 expression, may benefit from a novel therapeutic approach targeting this protein.

## Figures and Tables

**Figure 1 pharmaceuticals-17-01526-f001:**
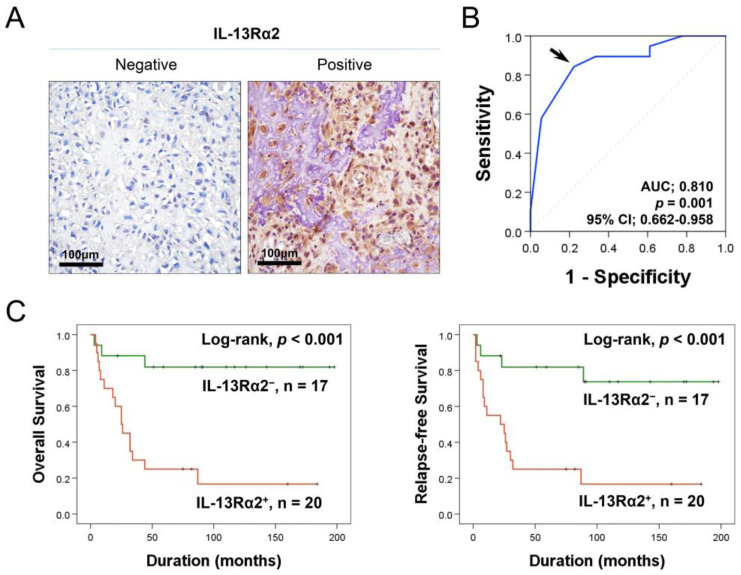
Immunohistochemical expression of IL-13Rα2 and survival analysis in 37 osteosarcomas. (**A**) IL-13Rα2 expression in osteosarcoma tissue, showing both positive and negative cases. (**B**) Analysis of the receiver operating characteristic curve to establish the threshold for immunohistochemical staining scores of IL-13Rα2. The selected threshold (arrow) corresponds to the point with the highest area under the curve (AUC), providing the best prediction of the death of patients from osteosarcoma. (**C**) Kaplan–Meier survival curves according to positivity of IL-13Rα2 expression for overall survival and relapse-free survival.

**Figure 2 pharmaceuticals-17-01526-f002:**
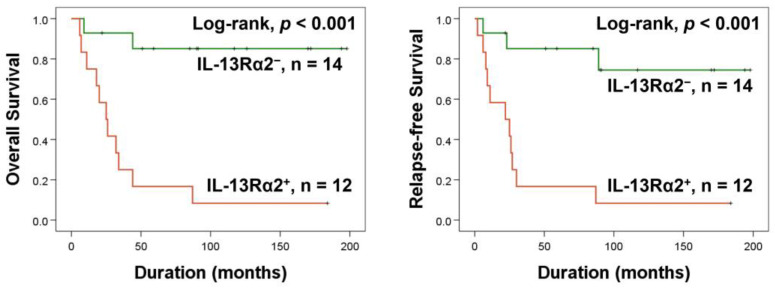
Kaplan–Meier survival analysis according to IL-13Rα2 expression in 26 osteosarcoma patients who received postoperative chemotherapy.

**Figure 3 pharmaceuticals-17-01526-f003:**
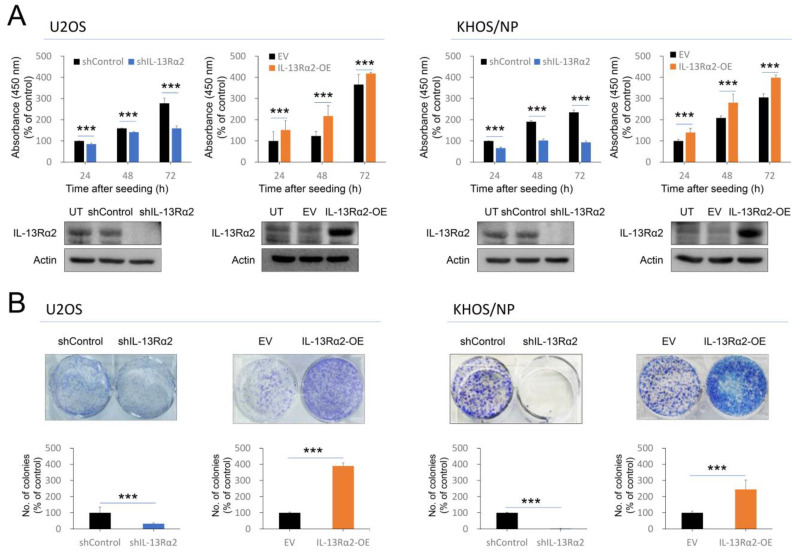
Effect of IL-13Rα2 on osteosarcoma cell proliferation: (**A**) U2OS and KHOS/NP cells were transfected with empty vector, shRNA for IL-13Rα2, or wild-type IL-13Rα2 plasmids and evaluated for the proliferation of cells with CCK-8 assay at 24 h, 48 h, and 72 h after seeding of 3 × 10^3^ cells in each well. The absorbance was measured at 450 nm wavelength. To evaluate the efficacy of transfection, Western blotting was performed for IL-13Rα2 and actin. (**B**) For a colony-forming assay, 3 × 10^3^ U2OS cells and 1 × 10^3^ KHOS/NP cells were seeded in 6-well plates and cultured for ten days. The number of colonies was counted using ColonyCountJ software (version 1.0). *** *p* < 0.001; EV, empty vector for wild-type IL-13Rα2; IL-13Rα2-OE, vector for wild-type IL-13Rα2; shControl, control vector for shRNA; shIL-13Rα2, vector for shRNA for IL-13Rα2; UT, untransfected cells.

**Figure 4 pharmaceuticals-17-01526-f004:**
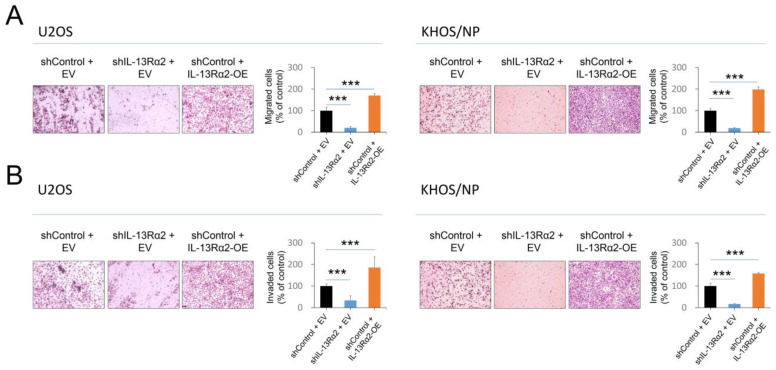
Effect of IL-13Rα2 on migration and invasion activity of osteosarcoma cells: (**A**) Migration assay was conducted by seeding 1 × 10^5^ cells into the upper chamber of each well. (**B**) Invasion assays were performed by seeding 2 × 10^5^ cells in each well of the Matrigel-coated upper chamber. Serum-free media was used in the upper chamber for both migration and invasion assays, while the lower chamber contained media with 10% serum to serve as a chemoattractant. The migrated or invaded cells were stained with Diff Quick staining solution and counted in five separate ×100 microscopic fields per well. Original magnification, ×400. *** *p* < 0.001; EV, empty vector for wild-type IL-13Rα2; IL-13Rα2-OE, vector for wild-type IL-13Rα2; shControl, control vector for shRNA; shIL-13Rα2, vector for shRNA for IL-13Rα2.

**Figure 5 pharmaceuticals-17-01526-f005:**
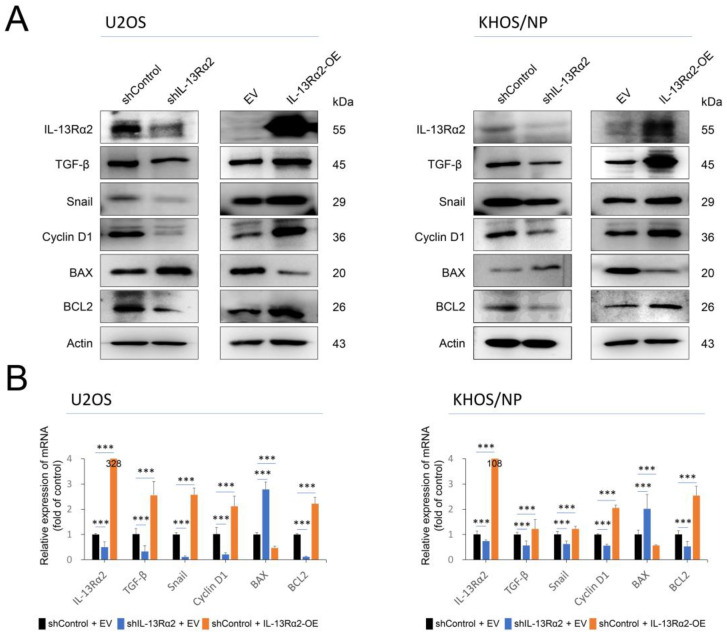
IL-13Rα2 is involved in the regulation of signaling molecules related to osteosarcoma cell proliferation and apoptosis: (**A**) Two osteosarcoma cells, U2OS and KHOS/NP, were transfected with empty vector, shRNA for IL-13Rα2, or wild-type IL-13Rα2 plasmids, and Western blotting performed for IL-13Rα2, TGF-β, snail, cyclin D1, BAX, BCL2, and actin. (**B**) Quantitative reverse-transcription polymerase chain reaction was performed for IL-13Rα2, TGF-β1, snail, cyclin D1, BAX, BCL2, and GAPDH in U2OS and KHOS/NP cells after transfection of vectors. The data were normalized with GAPDH expression. *** *p* < 0.001; EV, empty vector for wild-type IL-13Rα2; IL-13Rα2-OE, vector for wild-type IL-13Rα2; shControl, control vector for shRNA; shIL-13Rα2, vector for shRNA for IL-13Rα2.

**Figure 6 pharmaceuticals-17-01526-f006:**
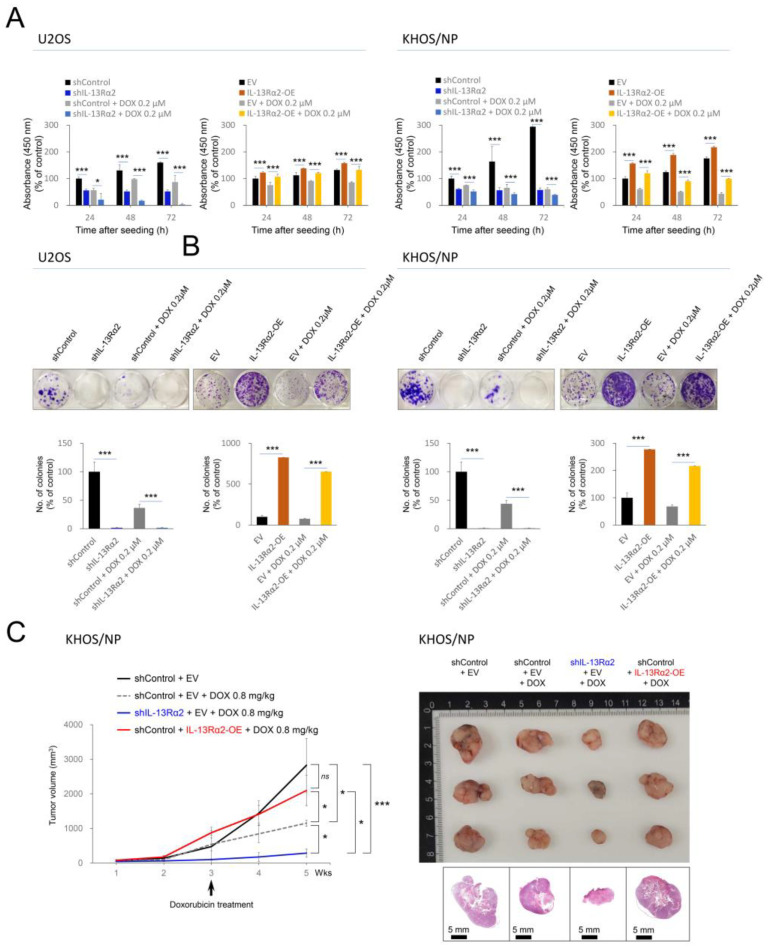
Effects of IL-13Rα2 on proliferation of osteosarcoma cells under treatment with doxorubicin: (**A**) U2OS and KHOS/NP cells transfected with empty vector, shRNA for IL-13Rα2, or wild-type IL-13Rα2 plasmids were treated with 0.2 μM doxorubicin. The proliferation of cells was measured with CCK-8 assay at 24 h, 48 h, and 72 h after seeding of 3 × 10^3^ cells. The absorbance was measured at 450 nm wavelength. (**B**) A colony-forming assay was conducted by seeding 3 × 10^3^ U2OS cells and 1 × 10^3^ KHOS/NP cells in 12-well plates and grown for seven days. The resulting colonies were stained with crystal violet and counted using ColonyCountJ software (version 1.0). (**C**) In vivo tumor growth was assessed in KHOS/NP cells transfected with empty vector, shRNA for IL-13Rα2, or wild-type IL-13Rα2 plasmids, along with analysis of the gross and histologic findings of resected tumors. Subcutaneous tumor implantation implants were established by injecting 2.5 × 10^6^ KHOS/NP cells into the back of mice. Three mice were used in each group. After three weeks, doxorubicin (4 mg/kg in dimethyl sulfoxide) was administered intraperitoneally once per week. Tumor volume was measured weekly using the formula length × width × height × 0.52. The mice were sacrificed two weeks after treatment with doxorubicin, and histologic sections were H&E stained. * *p* < 0.05; *** *p* < 0.001; *ns*, not significant; EV, empty vector for wild-type IL-13Rα2; IL-13Rα2-OE, vector for wild-type IL-13Rα2; shControl, control vector for shRNA; shIL-13Rα2, vector for shRNA for IL-13Rα2; DOX, doxorubicin.

**Figure 7 pharmaceuticals-17-01526-f007:**
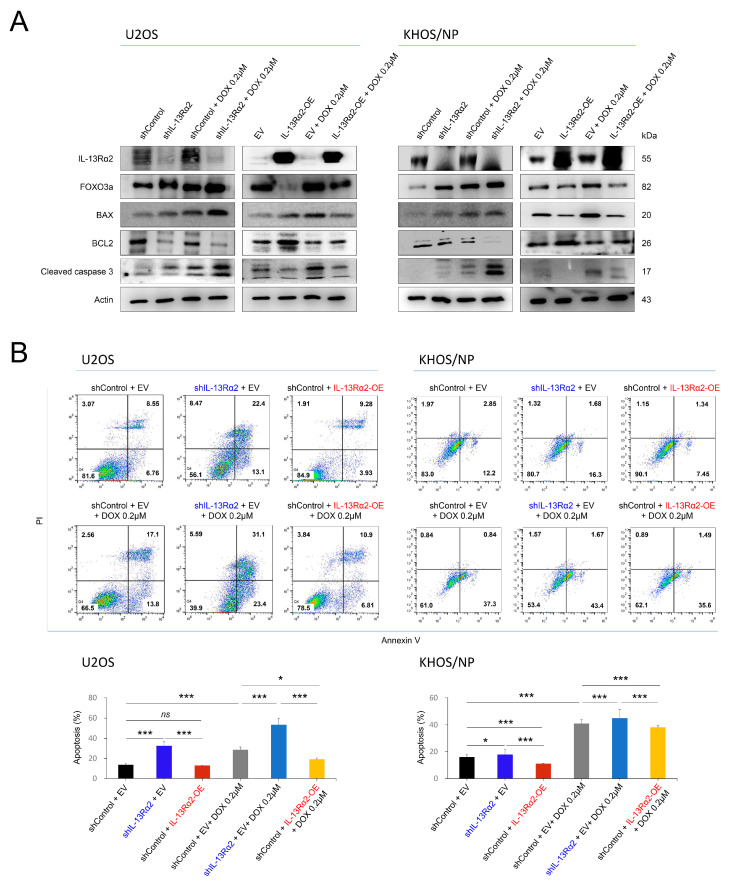
Effects of IL-13Rα2 on apoptosis of osteosarcoma cells under treatment with doxorubicin: (**A**) U2OS and KHOS/NP cells were transfected with vectors, followed by 0.2 μM doxorubicin treatment and Western blot for IL-13Rα2, FOXO3a, BAX, BCL2, cleaved caspase 3, and actin was performed. (**B**) Flow cytometric analysis for apoptosis of osteosarcoma cells under treatment with doxorubicin. U2OS and KHOS/NP cells transfected with vectors were treated with 0.2 μM doxorubicin for 24 h, stained with Annexin V and propidium iodide, and examined using flow cytometry. The right upper and right lower quadrants were considered apoptotic populations. * *p* < 0.05; *** *p* < 0.001; *ns*, not significant; EV, empty vector for wild-type IL-13Rα2; IL-13Rα2-OE, vector for wild-type IL-13Rα2; shControl, control vector for shRNA; shIL-13Rα2, vector for shRNA for IL-13Rα2; DOX, doxorubicin.

**Table 1 pharmaceuticals-17-01526-t001:** The association between the expression of IL-13Rα2 and the clinical characteristics of osteosarcoma in 37 osteosarcomas.

Characteristics		No.	IL-13Rα2	
			Positive	*p **
Age, years	<30	24	12 (50%)	0.501
	≥30	13	8 (62%)	
Sex	Male	25	17 (68%)	0.014
	Female	12	3 (25%)	
Tumor size	≤8 cm	19	8 (42%)	0.134
	>8 cm	18	12 (67%)	
Stage	I and II	26	12 (46%)	0.138
	III and IV	11	8 (73%)	
T category	1	17	7 (41%)	0.316
	2	16	10 (63%)	
	3 and 4	4	3 (75%)	
N category	N0	34	18 (53%)	0.647
	N1	3	2 (67%)	
M category	M0	29	14 (48%)	0.179
	M1	8	6 (75%)	

* *p*-values were calculated with a chi-squared test.

**Table 2 pharmaceuticals-17-01526-t002:** Univariate survival analysis in 37 osteosarcoma patients.

Characteristics	No.	OS		RFS	
		HR (95% CI)	*p*	HR (95% CI)	*p*
Age, years, ≥30 (vs. <30)	13/37	2.593 (1.047–6.419)	0.039	2.944 (1.217–7.123)	0.017
Sex, male (vs. female)	25/37	0.833 (0.299–2.323)	0.727	0.686 (0.248–1.899)	0.468
Tumor size, ≥8 cm (vs. <8 cm)	18/37	3.319 (0.253–8.790)	0.016	2.987 (1.179–7.564)	0.021
Stage, III and IV (vs. I and II)	11/37	3.115 (1.236–7.853)	0.016	3.652 (1.472–9.062)	0.005
T category, 1	17/37	1	0.059	1	0.020
2	16/37	3.309 (1.144–9.569)	0.027	3.383 (1.168–9.801)	0.025
3 and 4	4/37	3.998 (0.950–16.828)	0.059	6.014 (1.589–22.760)	0.008
N category, N1 (vs. N0)	3/37	4.031 (0.838–19.400)	0.082	2.636 (0.571–12.177)	0.214
M category, M1 (vs. M0)	8/37	3.812 (1.408–10.316)	0.008	3.282 (1.206–8.926)	0.020
IL-13Rα2, positive (vs. negative)	20/37	7.191 (2.063–25.067)	0.002	5.668 (1.848–17.385)	0.002

OS, overall survival; RFS, relapse-free survival; HR, hazard ratio; 95% CI, 95% confidence interval.

**Table 3 pharmaceuticals-17-01526-t003:** Multivariate survival analysis in 37 osteosarcoma patients.

Characteristics	OS		RFS	
	HR (95% CI)	*p*	HR (95% CI)	*p*
Age, years, ≥30 (vs. <30)			3.159 (1.267–7.873)	0.014
Stage, III and IV (vs. I and II)			2.865 (1.130–7.262)	0.027
IL-13Rα2, positive (vs. negative)	7.191 (2.063–25.067)	0.002	6.161 (1.909–19.883)	0.002

OS, overall survival; RFS, relapse-free survival; HR, hazard ratio; 95% CI, 95% confidence interval. The factors considered in multivariate analysis were age, tumor size, tumor stage, T category, M category, and IL-13Rα2 expression.

**Table 4 pharmaceuticals-17-01526-t004:** Univariate and multivariate survival analysis in 26 patients treated with chemotherapy after surgery.

Characteristics		No.	OS		RFS	
			HR (95% CI)	*p*	HR (95% CI)	*p*
Univariate analysis						
	IL-13Rα2, positive (vs. negative)	12/26	11.985 (2.600–55.254)	0.001	8.208 (2.203–30.583)	0.002
Multivariate analysis						
	Stage, III and IV (vs. I and II)				3.210 (1.052–9.789)	0.040
	IL-13Rα2, positive (vs. negative)		11.985 (2.600–55.254)	0.001	7.461 (1.946–28.605)	0.003

OS, overall survival; RFS, relapse-free survival; HR, hazard ratio; 95% CI, 95% confidence interval. The factors considered in multivariate analysis were age, tumor size, tumor stage, T category, M category, and IL-13Rα2 expression.

**Table 5 pharmaceuticals-17-01526-t005:** Primer sequences used for a quantitative reverse-transcription polymerase chain reaction.

Gene	Primer Sequence	Accession Number
*IL-13RA2* (IL-13Rα2)	Forward 5′-TGACGGAATTTGGAGTGAGTGG-3′ Reverse 5′-CCAAATGGTAGCCAGAAACGTAGC-3′	NM_000640.2
*TGFB1* (TGF-β1)	Forward 5′-CCCACAACGAAATCTATGACAA-3′ Reverse 5′-AAGATAACCACTCTGGCGAGTG-3′	NM_000660.7
*SNAL1* (snail)	Forward 5′-ACCCCACATCCTTCTCACTG-3′ Reverse 5′-TACAAAAACCCACGCAGACA-3′	NM_005985.3
*CCND1* (cyclin D1)	Forward 5′-GAGGAAGAGGAGGAGGAGGA-3′ Reverse 5′-GAGATGGAAGGGGGAAAGAG-3′	NM_053056.2
*BCL2*	Forward 5′-ATCGCCCTGTGGATGACTGAGT-3′ Reverse 5′-GCCAGGAGAAATCAAACAGAGGC-3′	NM_000633.3
*BAX*	Forward 5′-TGGCAGCTGACATGTTTTCTGAC-3′ Reverse 5′-TCACCCAACCACCCTGGTCTT-3′	NM_138761.4
*GAPDH*	Forward 5′-AACAGCGACACCCACTCCTC-3′ Reverse 5′-GGAGGGGAGATTCAGTGTGGT-3′	NM_001256799.1

## Data Availability

Data is contained within the article.

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
