# Peer review of "IL-13Rα2 Is Involved in Resistance to Doxorubicin and Survival of Osteosarcoma Patients"

_pharmaceuticals, 2024, doi:10.3390/ph17111526_

Round 1
Reviewer 1 Report
Comments and Suggestions for Authors
The manuscript submitted by Karamikheirabad and coworkers aims to investigate the diagnostic value of IL-13Rα2 in predicting survival in osteosarcoma as well as to describe the biological mechanisms induced by this receptor causing the tumorigenic phenotype. Since the IL-13 pathway is poorly investigated in osteosarcoma, the study potentially possesses a good level of novelty. Moreover, the experimental workflow is logic and consistent, moving from evidences in patients to the dissection of underlying molecular mechanisms, and the manuscript is overall well written. However, I detected some methodological problems that should be amended.
The major problem is related to the phenotype of cells transfected with control vectors (shControl and EV), which are too dissimilar each other while they should be overlapping or at least similar. This is evident starting from Western blots for IL-13Rα2. The two controls are not consistent each other, and this occurs in a similar way in both the cell lines (Figure 3 and 5). Indeed, when cells are transfected with shControl they show to express IL-13Rα2, but when cells are transfected with EV, IL-13Rα2 band becomes barely detectable. How is this possible, considering that in both the cases cells are transfected by using lipofectamine (therefore, the effect of different treatments on IL-13Rα2 can be excluded)? Moreover, the blot showing the basal expression (untransfected cells) is missing and should be added.
In addition, rates of proliferation seem to change based on the vector in U2OS cells. For both the cell lines, I noticed that graph scales are different between silencing and over-expression, while it would be far better to maintain the same scale, since they are in the same range. Anyway, while in KHOS/NP cells the absorbance of silencing and over-expression controls is very similar, in U2OS cells it shows marked differences, with over-expression controls displaying a higher absorbance compared to silencing controls. Just as an example, control absorbance at 72h in silencing experiments is around 1.8, while in over-expression experiment such absorbance is almost doubled, around 3. Since the number of cells is the same for silencing and over-expression experiments, what is the reason of such a difference?
In migration, invasiveness and mRNA quantization experiments, only EV has been chosen has control. Although it could sound reasonable, since the two controls should have similar behaviours, it is actually not, has already shown by IL-13Rα2 expression. This is reinforced by Western blot experiments, in which the two controls show marked differences for almost all the tested proteins. This means that the two controls are not interchangeable and the choice of using one or the other is arbitrary. Anyway, again the major problem here is related to controls, that shouldn’t show such discrepancies.
In Figure 6A, controls for KHOS/NP cells are quite dissimilar at both 24h and 48h. In Figure 6C, mice treated with shControl are missing. Although I understand that experiments on animals must be as limited as possible, the discrepancies induced by the different vectors do not justify the choice
Western blots in Figure 7 show the same problems mentioned above as well as the not justified choice of using EV only as control for Annexin-V experiments. Moreover, I recommend the author to recheck the compensation used for KHOS/NP cells, since it does not seem to be optimal (probably, there is still FITC signal in PI channel).
Minor revisions:
Table 1: I recommend to mention the statistical test used for the analysis at the end of the table (I guess it is Chi squared test, but it should be specified).
Line 137: I would suggest “proliferation of cells as measured with the CCK-8 assay”
Figure 3: as for panel A, I recommend to use the same scale for bar graphs in panel B, in order to better show that transfection controls display the same level of colonies development (I guess is around 100 for both the cells and both the controls). Accordingly, I would choose more representative plates for KHOS/NP controls (EV show more numerous colonies than shControl, while they should be similar as for U2OS).
IL-13 pathway (cytokine, receptors, producing cells) has been suggested to be involved in resistance to therapy in several types of cancer and in the context of different regimens (doi: 10.1155/2018/2606834; doi: 10.1002/cnr2.1701; doi: 10.3389/fimmu.2022.811131; doi: 10.1126/sciimmunol.abn0175; doi: 10.1038/s41418-023-01209-x. ). This suggests that induction of IL-13 might be a common mechanism of tumor resistance, an aspect that should be mentioned in the discussion
Line 273: it is “targeted”
Line 446: the sentence “Positivity for IL-13Rα2 immunohistochemical staining was determined using receiver operating characteristic curve analysis to predict the death of osteosarcoma patients” is not correct, since it is actual the opposite. I would suggest “Positivity for IL-13Rα2 immunohistochemical staining was used in receiver operating characteristic curve analysis to…”
Figure legends: the statistical test used for comparisons should be reported. Moreover, two asterisks usually correspond to a p-value < 0.01, while three asterisks are used for p-values < 0.001. Figures and/or legends should be corrected accordingly
Author Response
Response to reviewer 1
We thank the reviewer for these insightful comments.
Comments from Reviewer #1:
The manuscript submitted by Karamikheirabad and coworkers aims to investigate the diagnostic value of IL-13Rα2 in predicting survival in osteosarcoma as well as to describe the biological mechanisms induced by this receptor causing the tumorigenic phenotype. Since the IL-13 pathway is poorly investigated in osteosarcoma, the study potentially possesses a good level of novelty. Moreover, the experimental workflow is logic and consistent, moving from evidences in patients to the dissection of underlying molecular mechanisms, and the manuscript is overall well written. However, I detected some methodological problems that should be amended.
We thank the reviewer for these insightful comments.
The major problem is related to the phenotype of cells transfected with control vectors (shControl and EV), which are too dissimilar each other while they should be overlapping or at least similar. This is evident starting from Western blots for IL-13Rα2. The two controls are not consistent each other, and this occurs in a similar way in both the cell lines (Figure 3 and 5). Indeed, when cells are transfected with shControl they show to express IL-13Rα2, but when cells are transfected with EV, IL-13Rα2 band becomes barely detectable. How is this possible, considering that in both the cases cells are transfected by using lipofectamine (therefore, the effect of different treatments on IL-13Rα2 can be excluded)? Moreover, the blot showing the basal expression (untransfected cells) is missing and should be added.
We thank the reviewer for their comment and agree with the reviewer. As shown in Figure 5B, when we induced overexpression of IL-13Rα2, IL-13Rα2 mRNA expression exceeded control levels by over 100-fold. As a result, if we tried to match the band density between shControl and EV for WT IL-13Rα2, the IL-13Rα2-OE band would be overly dense and overlap with the control band. Thus, we focused on showing the relatively high IL-13Rα2 level in cells transfected with the wild-type IL-13Rα2. However, in response to the comment of the reviewer, we revised the western blot bands in Figure 3A to show the untransfected basal levels, shControl/EV, and shIL-13Rα2/IL-13Rα2-OE. In the revised figure, we tried to show bands for both basal level and shControl/EV groups.
In addition, rates of proliferation seem to change based on the vector in U2OS cells. For both the cell lines, I noticed that graph scales are different between silencing and over-expression, while it would be far better to maintain the same scale, since they are in the same range. Anyway, while in KHOS/NP cells the absorbance of silencing and over-expression controls is very similar, in U2OS cells it shows marked differences, with over-expression controls displaying a higher absorbance compared to silencing controls. Just as an example, control absorbance at 72h in silencing experiments is around 1.8, while in over-expression experiment such absorbance is almost doubled, around 3. Since the number of cells is the same for silencing and over-expression experiments, what is the reason of such a difference?
We thank the reviewer for this comment and agree with the reviewer. Although we summarized our results briefly in a single manuscript, this research was conducted over three years to elucidate the role of IL-13Rα2 in osteosarcoma. During this time, variations in cell conditions and transfection efficiency occurred across different experimental stages, leading to slight differences in results despite using the same cell types and quantities. Each experiment was conducted with its own control, which explains the use of controls in each instance. Thus, our focus was on showing the relative values in experimental groups (knockdown or overexpression) compared to the control group. Additionally, as noted by indicating the first two authors as having contributed equally, two primary authors conducted key experiments, which may have led to slight variations in the control's expression pattern. Nonetheless, we consistently aimed to show the relative expression values between control and experimental groups to illustrate the effects of IL-13Rα2 expression. In response to the reviewer's comment, we revised the figures to use consistent scales and present data as a percentage of the control.
In migration, invasiveness and mRNA quantization experiments, only EV has been chosen has control. Although it could sound reasonable, since the two controls should have similar behaviours, it is actually not, has already shown by IL-13Rα2 expression. This is reinforced by Western blot experiments, in which the two controls show marked differences for almost all the tested proteins. This means that the two controls are not interchangeable and the choice of using one or the other is arbitrary. Anyway, again the major problem here is related to controls, that shouldn’t show such discrepancies.
We thank the reviewer for their comment and apologize for any lack of clarity in our initial explanation. In this study, to show the effects of knockdown and overexpression in the same experiment, the label "EVs" was used in some samples to refer to cells transfected with both shControl and an empty vector for overexpression. Additionally, the group labeled "shControl" compared with the “EVs” group refers to cells transfected with both shRNA for IL-13Rα2 and an empty vector. The "IL-13Rα2-OE" group, when compared with the “EVs” group, consists of cells transfected with shControl and a vector for wild-type IL-13Rα2. In response to the reviewer’s comment, we clarified this point in the figure and revised it as follows: EVs; shControl + EV, shIL-13Rα2; shIL-13Rα2 + EV, IL-13Rα2-OE; shControl + IL-13Rα2-OE.
In Figure 6A, controls for KHOS/NP cells are quite dissimilar at both 24h and 48h. In Figure 6C, mice treated with shControl are missing. Although I understand that experiments on animals must be as limited as possible, the discrepancies induced by the different vectors do not justify the choice
We thank the reviewer for this comment and apologize once again for this point. As we respond to previous comments, in response to the comment of the reviewer, we have clarified this point in the figure and revised it as follows: EVs; shControl + EV, shIL-13Rα2; shIL-13Rα2 + EV, IL-13Rα2-OE; shControl + IL-13Rα2-OE.
Western blots in Figure 7 show the same problems mentioned above as well as the not justified choice of using EV only as control for Annexin-V experiments. Moreover, I recommend the author to recheck the compensation used for KHOS/NP cells, since it does not seem to be optimal (probably, there is still FITC signal in PI channel).
We thank the reviewer for this comment and apologize again for this point. As we respond to previous comments, in response to the comment of the reviewer, we clearly described this point in the figure and revised it as next: EVs; shControl + EV, shIL-13Rα2; shIL-13Rα2 + EV, IL-13Rα2-OE; shControl + IL-13Rα2-OE. In addition, in response to the comment of the reviewer, we repeated the flow cytometric apoptotic analysis and updated Figure 7B accordingly.
Minor revisions:
Table 1: I recommend to mention the statistical test used for the analysis at the end of the table (I guess it is Chi squared test, but it should be specified).
We thank the reviewer for this comment. In response to the comment of the reviewer, we mentioned the statistical test used for the analysis at the end of the Table 1.
Line 137: I would suggest “proliferation of cells as measured with the CCK-8 assay”
We thank the reviewer for this comment. In response to the comment of the reviewer, we have revised it.
Figure 3: as for panel A, I recommend to use the same scale for bar graphs in panel B, in order to better show that transfection controls display the same level of colonies development (I guess is around 100 for both the cells and both the controls). Accordingly, I would choose more representative plates for KHOS/NP controls (EV show more numerous colonies than shControl, while they should be similar as for U2OS).
We thank the reviewer for this comment. In response to the comment of the reviewer, we have revised Figure 2B to use the same scale for the bar graph and have changed it with a representative plate for KHOS/NP control.
IL-13 pathway (cytokine, receptors, producing cells) has been suggested to be involved in resistance to therapy in several types of cancer and in the context of different regimens (doi: 10.1155/2018/2606834; doi: 10.1002/cnr2.1701; doi: 10.3389/fimmu.2022.811131; doi: 10.1126/sciimmunol.abn0175; doi: 10.1038/s41418-023-01209-x. ). This suggests that induction of IL-13 might be a common mechanism of tumor resistance, an aspect that should be mentioned in the discussion.
We thank the reviewer for this comment. In response to the comment of the reviewer, concerning this point, we have revised the discussion section of the manuscript as follows:
Supportively, the IL-13 pathway has been suggested to be involved in resistance to therapy in several types of cancers. In NK/T-cell lymphoma cells, IL-13 induced resistance to Adriamycin by promoting the expression of ATP-binding cassette sub-family C member 4 [46]. Additionally, IL-4 and IL-13 produced by group 2 innate lymphoid cells suppressed antitumor immunity in a colorectal cancer model [47], and the production of IL-13 by group 2 innate lymphoid cells was upregulated by melanoma cells in vitro [48]. These findings suggest that a cross-talk between tumor cells and IL-13 production by innate lymphoid cells contributes to cancer progression. Moreover, blocking immune-suppressor-inducing molecules such as netrin-1 suppressed resistance to chemotherapy and immune checkpoint inhibitors in a breast cancer model [49]. Higher expression of IL-13Rα2 was also associated with castration resistance in prostate cancer cells [50]. Therefore, in this context, it is suggested that IL-13 induction might be a common mechanism of tumor resistance, with IL-13Rα2 contributing to resistance as the receptor for IL-13.
Line 273: it is “targeted”
We thank the reviewer for this comment. In response to the comment of the reviewer, we have revised it.
Line 446: the sentence “Positivity for IL-13Rα2 immunohistochemical staining was determined using receiver operating characteristic curve analysis to predict the death of osteosarcoma patients” is not correct, since it is actual the opposite. I would suggest “Positivity for IL-13Rα2 immunohistochemical staining was used in receiver operating characteristic curve analysis to…”
We thank the reviewer for this comment. In response to the comment of the reviewer, we have revised it.
Figure legends: the statistical test used for comparisons should be reported. Moreover, two asterisks usually correspond to a p-value < 0.01, while three asterisks are used for p-values < 0.001. Figures and/or legends should be corrected accordingly
We thank the reviewer for this comment. In response to the comment of the reviewer, we have revised it.

Reviewer 2 Report
Comments and Suggestions for Authors
Manuscript NO: pharmaceuticals-3263953
Manuscript Type: Research Article
Manuscript Title: IL-13Rα2 is Involved in Resistance to Doxorubicin and Survival of Osteosarcoma Patients
MAIN
This is an interesting study reporting significant data on the implication of IL-13Rα2 in osteosarcoma, being obtained ex vivo, with tumor isolated from patients, in vitro, with osteosarcoma cells and in vivo in mice. In this manuscript, the authors, Dr. Karamikheirabad and colleagues aimed in evaluating, for the first time, the role of IL-13Rα2 in osteosarcoma and paritcularly identified IL13Rα2 as a potential further prognostic indicator in osteosarcoma patients, while IL13Rα2 has also been reported to influences the resistance to the chemotherapeutic agent doxorubicin in osteosarcoma cells an in vivo. This work is well conducted and well written. The data are robust, clear and well discussed.
Below are my comments to further enhance the work.
1. To strengthen the rationale for selecting IL-13Rα2, I recommend including more details on the role of this protein in bone tissue in physiological conditions. Several studies have shown that IL-13 in general but also specifically IL-13Rα2 play a role in bone homeostasis and it can therefore be dysregulated in osteosarcoma. These novel notions will underscore the potential of IL-13Rα2 dyregulation in osteosarcoma. Several suggested works which deserve mention in this context
https://pubmed.ncbi.nlm.nih.gov/38291404/
https://www.sciencedirect.com/science/article/pii/S0021925820706617
https://www.thelancet.com/article/S2352-3964(19)30477-3/fulltext
2. Consistently, the sole fact that the studies on IL-13Rα2 in human osteosarcoma are limited is not a good rationale. Please improve the sentence before the aim at the end of the introduction
3. The role of IL-13Rα2 should also be investigated in a cell model of normal control cells, such as hFOB 1.19 (human fetal osteoblasts) or MC3T3-E1 (mouse pre-osteoblasts), to better understand its expression and function in healthy bone tissue. This comparison with non-cancerous cells is crucial for distinguishing tumor-specific dysregulation from normal physiological processes. The absence of such an analysis should be acknowledged as a limitation of the study and addressed in the discussion, as it could provide valuable insights into whether IL-13Rα2 represents a viable therapeutic target specific to osteosarcoma other than a prognostic marker
BELOW SOME MINOR COMMENT
3. Abstract, the objective of the study shold be stated under the Background/Objectives sub section
4. Introduction, please incude these recently published reviews on osteosarcoma https://www.frontiersin.org/journals/cell-and-developmental-biology/articles/10.3389/fcell.2024.1394339/full, https://pubmed.ncbi.nlm.nih.gov/38677541/, https://pubmed.ncbi.nlm.nih.gov/39245737/
5. If possible, I suggest including more references in the methods. These two paeprs should be included for WB analyses DOI:10.3389/fonc.2021.679285 and Sci Rep . 2015 Oct 8:5:14983. doi: 10.1038/srep14983.
6. Methods, line 388 “2x105” , 5 should be uppercase
1. Abstract, better “tumor tissues from 37 human osteosarcomas”, line 21
2. Discussion, lines 318-319 the specific tumor model shold be detailed
Author Response
Response to reviewer 2
We thank the reviewer for these insightful comments.
Comments from Reviewer #2:
MAIN
This is an interesting study reporting significant data on the implication of IL-13Rα2 in osteosarcoma, being obtained ex vivo, with tumor isolated from patients, in vitro, with osteosarcoma cells and in vivo in mice. In this manuscript, the authors, Dr. Karamikheirabad and colleagues aimed in evaluating, for the first time, the role of IL-13Rα2 in osteosarcoma and particularly identified IL13Rα2 as a potential further prognostic indicator in osteosarcoma patients, while IL13Rα2 has also been reported to influences the resistance to the chemotherapeutic agent doxorubicin in osteosarcoma cells an in vivo. This work is well conducted and well written. The data are robust, clear and well discussed.
We thank the reviewer for these insightful comments.
Below are my comments to further enhance the work.
- To strengthen the rationale for selecting IL-13Rα2, I recommend including more details on the role of this protein in bone tissue in physiological conditions. Several studies have shown that IL-13 in general but also specifically IL-13Rα2 play a role in bone homeostasis and it can therefore be dysregulated in osteosarcoma. These novel notions will underscore the potential of IL-13Rα2 dysregulation in osteosarcoma. Several suggested works which deserve mention in this context: https://pubmed.ncbi.nlm.nih.gov/38291404/,
https://www.sciencedirect.com/science/article/pii/S0021925820706617
https://www.thelancet.com/article/S2352-3964(19)30477-3/fulltext
We thank the reviewer for this comment. In response to the comment of the reviewer, concerning this point, we have revised the manuscript as follows:
In addition, IL-13 is involved in the regulation of osteoblast and osteoclast activation [35,36], and IL-13Rα2 regulates osteoclastic differentiation through the mitogen-activated protein kinase/Akt pathway [37]. Therefore, considering the potential roles of IL-13/IL-13Rα2 pathway in osteogenesis and its roles in the development and progression of osteosarcoma, this study aimed to investigate the expression and roles of IL-13Rα2 in the progression of osteosarcoma by evaluating the expression of IL-13Rα2 in human osteosarcoma tissue samples and cell lines.
- Consistently, the sole fact that the studies on IL-13Rα2 in human osteosarcoma are limited is not a good rationale. Please improve the sentence before the aim at the end of the introduction
We thank the reviewer for this comment. In response to the comment of the reviewer, concerning this point, we have revised the manuscript as follows:
In addition, IL-13 is involved in the regulation of osteoblast and osteoclast activation [35,36], and IL-13Rα2 regulates osteoclastic differentiation through the mitogen-activated protein kinase/Akt pathway [37]. Therefore, considering the potential roles of IL-13/IL-13Rα2 pathway in osteogenesis and its roles in the development and progression of osteosarcoma, this study aimed to investigate the expression and roles of IL-13Rα2 in the progression of osteosarcoma by evaluating the expression of IL-13Rα2 in human osteosarcoma tissue samples and cell lines.
- The role of IL-13Rα2 should also be investigated in a cell model of normal control cells, such as hFOB 1.19 (human fetal osteoblasts) or MC3T3-E1 (mouse pre-osteoblasts), to better understand its expression and function in healthy bone tissue. This comparison with non-cancerous cells is crucial for distinguishing tumor-specific dysregulation from normal physiological processes. The absence of such an analysis should be acknowledged as a limitation of the study and addressed in the discussion, as it could provide valuable insights into whether IL-13Rα2 represents a viable therapeutic target specific to osteosarcoma other than a prognostic marker
We thank the reviewer for this comment. In response to the comment of the reviewer, concerning this point, we have revised the manuscript as follows:
In addition, when considering the role of IL-13Rα2 in bone homeostasis and its dysregulation in osteosarcoma, comparing it with non-cancerous cells is essential to distinguish tumor-specific dysregulation from normal physiological processes. This distinction is crucial to employ IL-13Rα2 as a therapeutic target of osteosarcoma. However, this study has a limitation in that we did not investigate the role of IL-13Rα2 in normal control osteoblastic cells. Therefore, further study is needed to provide val-uable insights into whether IL-13Rα2 represents a viable, specific therapeutic target for osteosarcoma, rather than solely as a prognostic marker.
BELOW SOME MINOR COMMENT
- Abstract, the objective of the study should be stated under the Background/Objectives sub section
We thank the reviewer for this comment. In response to the comment of the reviewer, we have revised Abstract as follow:
Therefore, this study aimed to investigate the expression and roles of IL-13Rα2 in the progression of osteosarcoma
- 4. Introduction, please include these recently published reviews on osteosarcoma https://www.frontiersin.org/journals/cell-and-developmental-biology/articles/10.3389/fcell.2024.1394339/full, https://pubmed.ncbi.nlm.nih.gov/38677541/, https://pubmed.ncbi.nlm.nih.gov/39245737/
We thank the reviewer for this comment. In response to the comment of the reviewer, we have revised Introduction section as follows:
Therefore, new therapeutic strategies for osteosarcoma need to be developed [30]. Recently, immunotherapeutic approaches, including immune checkpoint inhibitors, tumor microenvironment modulators, cytokine therapy, and chimeric antigen receptor T-cell therapy, are under evaluation in the treatment of osteosarcoma [31,32].
- If possible, I suggest including more references in the methods. These two papers should be included for WB analyses DOI:10.3389/fonc.2021.679285 and Sci Rep . 2015 Oct 8:5:14983. doi: 10.1038/srep14983.
We thank the reviewer for this comment. In response to the comment of the reviewer, we have cited suggested references.
- Methods, line 388 “2x105”, 5 should be uppercase
- Abstract, better “tumor tissues from 37 human osteosarcomas”, line 21
- Discussion, lines 318-319 the specific tumor model should be detailed
We thank the reviewer for this comment. In response to the comment of the reviewer, we have corrected the typographical error and specified the tumor model in the relevant sentence within the discussion section of the manuscript.
